# Combining the Broadband Coverage and Speed to Improve Fiscal System Efficiency in the Eastern European Union Countries

**Marius Dalian Doran** [1], **Silvia Puiu** [2,*], **Dorel Berceanu** [3], **Alexandra Mădălina Țăran** [1], **Iulia Para** [4] and **Jenica Popescu** [3]

1. Doctoral School of Economics and Business Administration, Faculty of Economics and Business Administration, West University of Timisoara, 300223 Timisoara, Romania
2. Department of Management, Marketing and Business Administration, Faculty of Economics and Business Administration, University of Craiova, 200585 Craiova, Romania
3. Department of Finance, Banking and Economic Analysis, Faculty of Economics and Business Administration, University of Craiova, 200585 Craiova, Romania
4. Department of Marketing and Economic International Relations, Faculty of Economics and Business Administration, West University of Timisoara, 300223 Timisoara, Romania
* Correspondence: silvia.puiu@edu.ucv.ro

**Abstract:** Current challenges triggered by the limited interactions between people and institutions during the pandemic crisis have emphasized the need to strengthen the digitization process of all public services. In this sense, we considered it opportune to carry out research in order to establish the impact of the technological infrastructure, in terms of coverage and download speed, on the efficiency of the fiscal policy expressed by the volume of income from taxes and fees. Therefore, we propose a robust regression model tested with S-estimator that allows for outliers in the dataset. The results indicate that an improvement in download speed has a significant positive effect on the level of tax collection and that a better broadband coverage improves the amount of revenues from taxes and contributions. In the analyzed countries, the technological infrastructure is developing, and the decision-makers should make efforts to reach the targets proposed in the Digital Agenda for Europe in terms of broadband coverage and speed.

**Keywords:** taxation; broadband coverage; download speed; tax collection

## 1. Introduction

The process of economic digitization remains one of the main points of interest in the agenda of the European Commission, and the development of the technological infrastructure related to a high-performance realization of the digitization process has set exact targets in the Digital Agenda for Europe (DAE).

There are many works that emphasize the importance of digitalization on the economy in general [1–4], but also in connection with taxation [5–8]. According to International Telecommunication Union [1], the developing countries should invest more in creating a better digital infrastructure, as stated in the report appreciating that mobile broadband is "the main digital technology contributing to economic development". Ivanov et al. [3] highlighted that Industry 4.0 in a country leads to a higher level of economic development. Moreover, Jiao and Sun [2] stated that digitalization leads to "urban economic growth". In addition to the impact it has on the economic growth, digitalization helps with a better collection of taxes, as emphasized by Tsindeliani et al. [5]. Furthermore, the authors [5] refer to a decrease in the "administrative burden" for the companies implementing digital technologies. Both reports of OECD [6,8] refer to the important role played by digitalization for the tax system of a country, facilitating the collection of higher

revenues and also leading to economic growth at both microeconomic and macroeconomic levels. Companies and governments thrive in a more digitalized world even if there are some challenges, such as cybersecurity, reluctance to change or lack of digital skills that should be tackled first.

In recent years, there has been an increased focus on the efforts of national economies to develop a digital infrastructure which brings important benefits for the economic development of a country. To capitalize all the advantages of a more digital economy, governments should ensure that their fiscal policies are established around digitalization, taking into account that this can contribute to a higher level of tax collection for the national budget.

For digitalization, we considered both broadband coverage and speed as variables that can influence the level of revenues from tax collection. The novelty of our research consists in developing a model in which we analyze the relationship between these two variables and the efficiency of fiscal systems in eastern European Union (EU) countries. Therefore, we started with the most recent report of the European Commission [9] regarding broadband coverage and speed and their evolution in the last years. For broadband coverage, we considered both fixed and mobile coverage since they might have a different impact on the way population and companies pay their taxes.

In addition to coverage, broadband speed is important, with the EU establishing a few objectives to be a "Gigabit society" until the year 2025 [10]. One of these objectives would be to have a broadband speed of at least 100 Mbps in all European households. The EU report [10] mentioned that the majority of households had in 2019 a speed higher than 30 Mbps, but there are many disparities between the EU countries and between urban and rural regions.

The benefits of having a digital economy with high coverage and speed of the Internet can be seen at both the microeconomic level (economic entities, individuals, families, institutions) and macroeconomic level. The opportunities provided at the microeconomic level can ensure better compliance in paying taxes and fees to central and local authorities, leading indirectly to more performance budgets and a more efficient fiscal system. The results of our study are helpful to managers in the public sector, who are involved in establishing sound fiscal policies that should consider the important role of digitalization in tax collection performance.

Even if many studies are searching for a direct relationship between digitalization and fiscal policy performance, the subject of the impact of technological infrastructure expressed in terms of coverage and speed on the level of tax collection has never been considered before in the literature. The research aims to analyze the impact of broadband coverage and speed in eastern EU countries on the fiscal efficiency of these countries. Both broadband coverage and speed are essential elements of digitalization. Therefore, we can highlight the novelty of the present study that aims to approach the impact of digitalization through broadband coverage and speed on the collection level in six eastern EU member states.

## 2. Literature Review

Many researchers studied the benefits of digitalization and its impact on the economy [2,3,11–13], but no research was carried out on the specific impact of broadband coverage and speed on the fiscal efficiency of a country. Rohman and Bohlin [11] analyzed the relationship between broadband speed and economic growth, proving a direct and positive relationship between these two variables. Hasbi and Bohlin [12] studied the influence of broadband speed on "income and unemployment", but the results varied in accordance with the job type and the city dimension.

Even if there are no papers focusing on the impact of broadband coverage and speed on fiscal efficiency, these variables are an indicator of digitalization which is extensively studied. The relationship between digitalization and fiscal performance has been

analyzed by several authors [5,7,14–19]. Koniagina [14] emphasized the role played by "digital technology" in raising tax revenues.

Gupta et al. [15] provided examples of good practices from Estonia and India, showing the impact of digitalization in the public sector on the usual individual who wants to pay their taxes. However, as the authors argued, for the benefits to be observed, there is a need for robust and courageous institutions that should establish and implement public policies even if, at first, the population might be reluctant to comply.

Vuković [16] highlighted that each country is unique, and many factors influence the success of digitalization in the public sector, mainly regarding the extent of tax collection and the level of compliance among the population. Hanrahan [17] analyzed the relationship between digitalization and tax collection, stating that this connection is not significantly studied by other researchers who focus more on the impact on economic growth rather than fiscal performance.

Ubago Martínez et al. [20] (p. 293) studied the efficiency of the fiscal system among OECD states taking into account multiple variables, such as "decentralization, simplification, digitalization, and education". The authors notice a higher level of taxes collected by the countries characterized by digitalization and decentralization.

Taking into account the availability of the data in EU member states [9] regarding broadband coverage and speed, we consider that it is essential to analyze the way these indicators, such as fixed broadband coverage, next-generation access (NGA) coverage, and long-term evolution (LTE) for mobile connection, might influence the way that the national administration prepares to collect taxes through a digital infrastructure from both the population and organizations.

There is an essential gap between rural and urban regions, but the report shows an improvement in 2021 compared with 2020 for rural areas. Therefore, fixed broadband coverage increased from 89.6% to 91.5% and the NGA coverage from 59.9% to 67.5%. The share for NGA considering the region is significantly higher in 2021 (90.1%), which shows important regional differences. These differences can also lead to a gap in tax collection and, in the end, higher disparities between these regions. Regarding LTE, according to the report [9], the LTE coverage is very high (above 99%), regardless of the region (urban or rural).

Analyzing the impact of broadband coverage and its indicators on fiscal performance is useful for public managers in tax administration who might develop and implement better strategies to improve tax collection. Therefore, in rural areas, the strategies might differ from those in big cities. As Ubago Martinez et al. [20] showed, education is another variable influencing tax collection, thus fiscal policies should be adjusted, considering all these factors.

The majority of researchers consider the impact of digitalization on economic growth, but the opposite is also true, and we notice this aspect from the differences in digitalization in rural and urban areas, especially when considering NGA. The level of economic development is lower in rural areas, thus many administrations lack the funds to implement digital tools or face the citizens' reluctance to change.

Rohman and Bohlin [21] studied the impact of broadband speed on the income of individuals in OECD countries and showed a direct relationship between these variables. Therefore, the authors highlighted a significant increase in income for those who benefit from a higher broadband speed. This is explained by jobs which require greater skills and are paid better. A higher broadband speed indicates that you can work more efficiently in a shorter period. Taking into account these results, we can understand the connection between higher household incomes and higher tax revenues collected by public administration.

Studies in the literature that analyze the impact of broadband coverage and speed on fiscal efficiency motivated us to develop a model which connects these variables. For the economic development of a country and also in times of crisis, such as the recent ones,

public administrators must better manage the funds they have and make all the efforts to raise tax collection in their communities [22–24].

## 3. Materials and Methods

### 3.1. Datasets and Variables' Selection

As stated above, the main purpose of this research is to identify the influence of the technological infrastructure in terms of coverage and speed on the level of fiscal revenues in the six eastern EU member states (Bulgaria, Czechia, Hungary, Poland, Romania, and Slovakia). The statistical data we used were collected from the official website of the European Commission for the period 2013–2021.

To quantify broadband speed, we used two indicators: Broadband coverage with speeds faster than 30 Mbps (BC30) and broadband coverage with speeds faster than 100 Mbps (BC100). For the coverage infrastructure, three variables were used: Fixed broadband coverage (FBC), which includes all the main fixed-line broadband access technologies; next-generation access (NGA) technologies, which include fixed-line broadband access technologies capable of achieving download speeds that meet the Digital Agenda objective of at least 30 Mbps coverage; and long-term evolution (LTE), which is the next-generation mobile service and supports peak downstream speeds of up to 100 Mbps. The dependent variable considered in the proposed analysis is represented by the total receipts from taxes and social contributions after deducting amounts assessed, but unlikely, to be collected (REV).

To analyze the descriptive statistics of the variables included in the model, it was necessary to use the raw data of the variables, not the transformed data. The descriptive statistics of the variables included in the research and presented in Table 1 have the purpose of measuring central tendency (mean and median), dispersion (standard deviation), and normality (Skewness and Kurtosis) of selected data [25]. Analyzing the Kurtosis normality indicator, we can see that for the variables REV and LTE, we have a leptokurtic distribution; for the other variables, we have platykurtic distribution, which is very close to normality. Regarding the skewness normality indicator, it can be easily observed that the distribution is asymmetric around its mean, all the independent variables present a long left tail (given by the negative values of the skewness) and the dependent variables REV present a long right tail. The values of the Jarque-Bera test and the associated probabilities for each of the variables confirm the null hypothesis of a normal distribution.

**Table 1.** Variables presentation and descriptive statistics.

|  | REV | BC30 | BC100 | FBC | NGA | LTE |
|---|---|---|---|---|---|---|
| Unit of measure | Million Euro | % of total households | % of total households | % of total households | % of total households | % of total households |
| Period | 2013–2021 | 2013–2021 | 2013–2021 | 2013–2021 | 2013–2021 | 2013–2021 |
| Mean | 59520.89 | 0.698490 | 0.518653 | 0.914776 | 0.737531 | 0.823898 |
| Median | 45503.70 | 0.724000 | 0.559000 | 0.944000 | 0.741000 | 0.963000 |
| Maximum | 187334.8 | 0.969000 | 0.864000 | 0.998000 | 0.919000 | 1.000000 |
| Minimum | 11831.30 | 0.345000 | 0.020000 | 0.791000 | 0.471000 | 0.000000 |
| Std. Dev. | 47302.91 | 0.142306 | 0.224065 | 0.058329 | 0.112102 | 0.267127 |
| Skewness | 1.479370 | −0.332334 | −0.417942 | −0.602276 | −0.397687 | −1.598545 |
| Kurtosis | 4.185545 | 2.428024 | 2.163486 | 2.243384 | 2.702390 | 4.441862 |
| Jarque-Bera | 20.74263 | 1.569920 | 2.855184 | 4.131130 | 1.472430 | 25.11322 |
| Probability | 0.000031 | 0.456138 | 0.239886 | 0.126747 | 0.478923 | 0.000004 |
| Sum | 2916524. | 34.22600 | 25.41400 | 44.82400 | 36.13900 | 40.37100 |
| Sum Sq. Dev. | $1.07 \times 10^{11}$ | 0.972048 | 2.409855 | 0.163309 | 0.603204 | 3.425136 |

*3.2. Research Design*

Taking into account that our datasets contain a large amount of outliers, the least square regression is not suitable for our analysis, and it is recommended to use robust regression to overcome the influence of extreme observation [26–28]. The mathematical form of the model equation is given as follows:

$$REV_t = \beta_1 + \beta_2 BC30_t + \beta_3 BC100_t + \beta_4 FBC_t + \beta_5 NGA_t + \beta_6 LTE_t + \varepsilon_t \tag{1}$$

where

REV is the amount of total receipts from taxes and social contributions;
BC30 is the broadband coverage with a speed faster than 30 Mbps;
BC100 is the broadband coverage with a speed faster than 100 Mbps;
FBC is the fixed broadband coverage;
NGA denote the next-generation access technologies;
LTE is the long-term evolution;
$\beta_{1..6}$ denote the associated coefficients of the variables;
t is the time period;
$\varepsilon$ is the standard error of regression.

The robust regression model allows for three estimation methods for computing the covariance matrix of the coefficient estimates [29–34]. In our research, we used the S-estimator (scale statistic estimator) introduced by Rousseeuw and Yohai [35]. The S-estimator is a member of the class of high-breakdown-value estimators [36]. This estimator finds the $\beta$ coefficients that provide the smallest values of the scale S, as follows:

$$\frac{1}{N-k} \sum_{i=1}^{N} h_0 \left( \frac{r_i(\beta)}{S} \right) = b \tag{2}$$

where

$r$ is the residual function given by $r_i(\beta) = r_i = y_i - X_i'\,\beta$;
$k$ is the coefficient vector;
$h_0()$ is a function with the tuning constant $c > 0$.

Calculation of S-estimator is computationally intensive, and there is a number of fast algorithms that provide accurate approximations [37]. We used the Fast-S algorithm of Salibian-Barrera and Yohai [38]. According to many authors, this method is appropriate for the type of analysis that we conducted [39–43]. Before proceeding to the regression analysis, it was necessary to perform unit root tests for all variables (Levin, Lin, Chu—LLC) [44], as well as a correlation matrix to detect the possible autocorrelation between the variables [45]. A Granger causality test [46,47] is applied to detect the direction of causality between the variables included in the model.

Based on the available dataset and the research method used, to answer the proposed objective, we considered it necessary to formulate and test the following hypotheses:

**Hypothesis 1 (H1).** *As part of the states that recently joined the European Union, in the six analyzed countries, the technological infrastructure is one that is developing.*

**Hypothesis 2 (H2).** *Broadband coverage speed has a significant impact on the level of collection of taxes and fees.*

**Hypothesis 3 (H3).** *Broadband coverage positively influences the level of fiscal revenues of central governments.*

## 4. Results and Discussion

### 4.1. Evolution of Technology Infrastructure in Terms of Speed and Coverage

In 2010, DAE was created as one of the flagship initiatives of the Europe 2020 strategy and included specific broadband coverage targets stretching to 2020: Universal broadband coverage of speed of at least 30 Mbps by 2020 and 50% of households should have broadband subscriptions of 100 Mbps or more by 2020 [9]. Examining coverage levels by the individual speed categories, at the end of 2021, it can be observed in Table 2 and Figure 1 that four countries from the eastern EU member states, namely Bulgaria, Czechia, Hungary, and Romania, exceeded the EU-27 average percentage of 89.9% for the fixed broadband networks, which are capable of providing them with actual download speed of at least 30 Mbps.

**Table 2.** Fixed broadband networks coverage by speed in 2021.

| Country | Romania | Bulgaria | Czechia | Hungary | Poland | Slovakia |
|---------|---------|----------|---------|---------|--------|----------|
| BC30 | 93.7% | 93.3% | 98.1% | 94.9% | 77.0% | 82.3% |
| BC100 | 88.6% | 91.9% | 89.2% | 88.7% | 69.2% | 75.4% |

For the other two countries, the values of the BC30 indicator are significantly below the European average, respectively 82.3% for Slovakia and 77% for Poland. This evolution was driven by the recorded growth in NGA coverage and the technological advancement provided by a higher number of very high-speed digital subscriber line (VDSL) networks capable of supporting a 30 Mbps download speed.

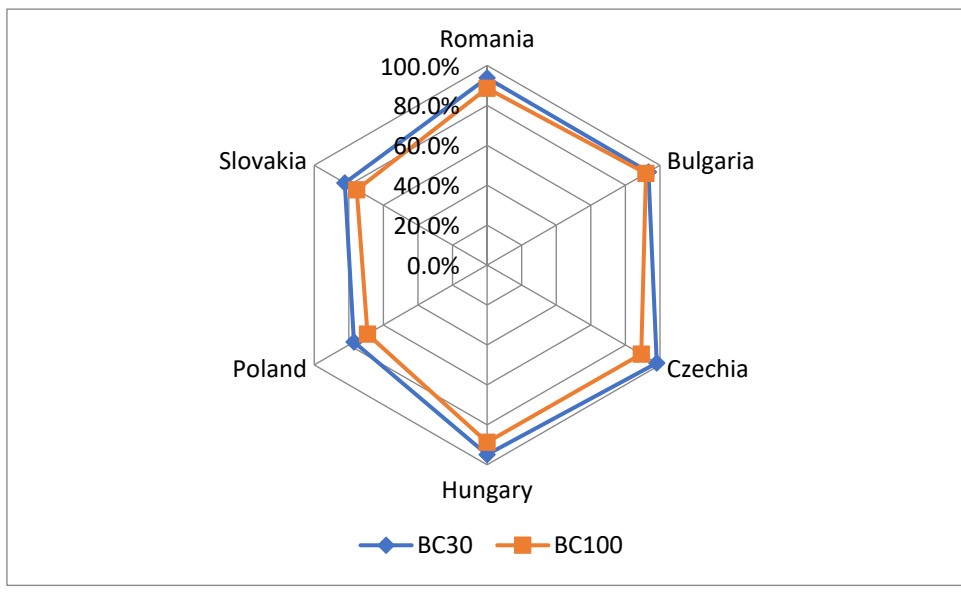

**Figure 1.** Fixed broadband networks coverage by speed in 2021.

Coverage of networks supporting at least 100 Mbps at the EU-27 level was 82.1% at the end of 2021. In Figure 2, it can be observed that Romania, Czechia, Bulgaria, and Hungary exceeded this percentage, and Poland (69.2%) and Slovakia (75.4%) were below this value. This is a result of the growth of two broadband types: VDSL2 vectoring coverage (connection to a VDSL2-enabled cabinet or exchange, and the vectoring solution is applied to receive at least 100 Mbps of download speed) and fiber to the premises (FTTP) coverage (connection to a fiber service without requiring the construction of new fiber infrastructure, which is available for connection within reasonable time and cost limits).

The overall fixed broadband coverage category has been designed to provide a measure of progress in the deployment of fixed broadband access technologies, which are capable of providing households with broadband services of at least 2 Mbps of download

speed. Four technologies make up the overall fixed broadband coverage: Digital subscriber line (DSL), cable, FTTP, and fixed wireless access (FWA).

In the year 2021, the European average of households with an FBC connection was 97.9%. Referring to this average, it can be seen in Table 3 that only two countries exceed this value, namely, the Czech Republic (99.9%) and Hungary (98.4%). Slovakia and Bulgaria are close to the European average with a percentage of over 97% of households with a fixed connection, while Romania (94.1%) and Poland (89.7%) are making considerable efforts to reach the targets proposed within DAE.

**Table 3.** Broadband networks coverage in 2021.

| Country | Romania | Bulgaria | Czechia | Hungary | Poland | Slovakia |
|---------|---------|----------|---------|---------|--------|----------|
| FBC | 94.1% | 97.3% | 99.9% | 98.4% | 89.7% | 97.4% |
| NGA | 93.3% | 93.3% | 92.6% | 96.7% | 78.2% | 84.3% |
| LTE | 99.9% | 99.9% | 99.8% | 99.7% | 99.9% | 98.4% |

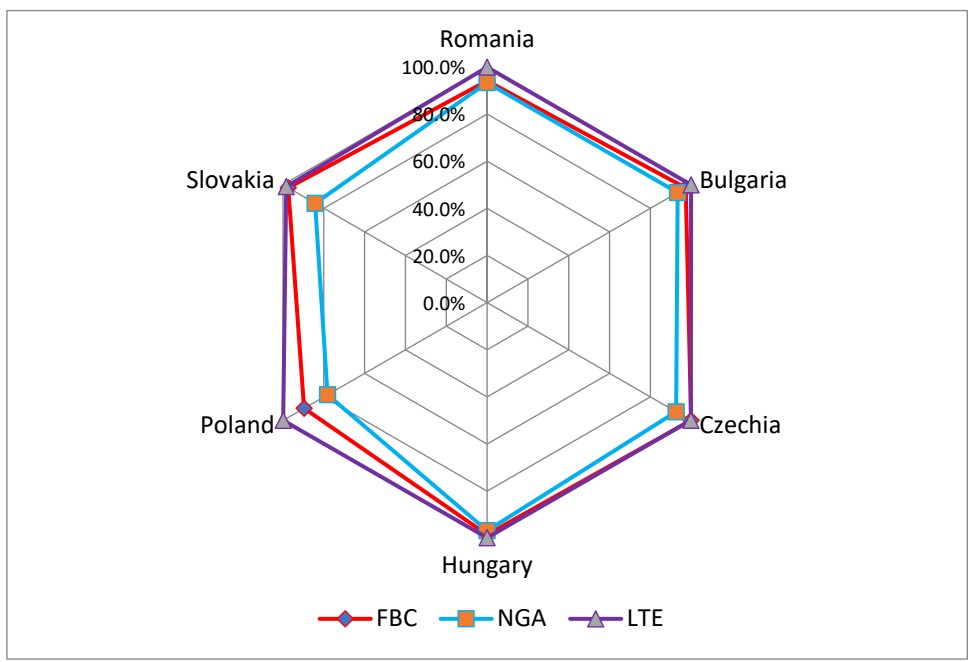

**Figure 2.** Broadband networks coverage in 2021.

The NGA category comprises technologies capable of delivering a service speed of at least 30 Mbps. This category must be improved considering the main objective of DAE to have complete coverage of European households at this speed by 2020. In Figure 2, it can be observed that this coverage registered the lowest values in the analyzed countries. Therefore, the analysis of the combination category constitutes an evaluation of the rollout of the relevant technologies and progress toward this goal. The European average for NGA is 90.1%. Of the six countries included in this research, only two are below this value (Slovakia—84.3% and Poland—78.2%).

The average LTE coverage metric is an important indicator since it has also been included as one of the components of the Connectivity dimension of the Digital Economy and Society Index (DESI) [48,49]. The European average of mobile broadband technologies coverage exceeds the percentage of 99.8% of all households. Of the six countries included in the analysis, only Slovakia (98.4%) is below this value and Hungary which is only 0.01 percentage points away from reaching the European average.

The statistical data analysis regarding the level of broadband coverage and the download speed confirms the first hypothesis formulated, H1, regarding the fact that in at least

two of the analyzed states (Poland and Slovakia), the technological infrastructure requires improvements to reach the targets established within the DAE.

### 4.2. Contribution of Broadband Technologies to Tax Collection

Figure 3 points out the relationship between total receipts from taxes and social contributions (REV) on the horizontal axis and broadband technologies indicators on the vertical axis. It indicates that the relationship is positive: More coverage and broadband speed lead to more revenues from taxes and contributions to the central government. Furthermore, the figure proves that the relationship is not linear. From the graph, we can observe two areas of concentration, one denser in the value range from EUR 10,000 to 80,000 million and another less dense in the value range from EUR 120,000 to 180,000 million.

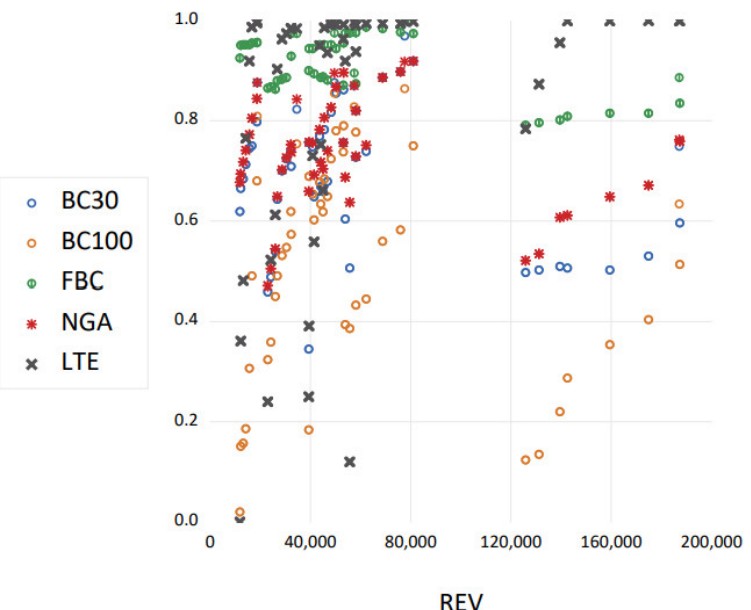

**Figure 3.** Broadband networks coverage and the total receipts from taxes and social contributions.

Figure 4 shows the presence of outliers in the datasets of our variables. As we mentioned in the previous section, it is necessary to identify these outliers, and if they are many, it is recommended to use robust regression for a more robust analysis [50]. As it can be seen, all the variables in the model present one or more outliers. Even if there are remedial measures for influential outliers and non-normal distributions, they are not always effective for large amounts of contamination and are not easy to automate. Occasionally, it is essential to keep outliers in the data and not remove them completely. Removing data will reduce the sample size, which is not good from an estimation point-of-view.

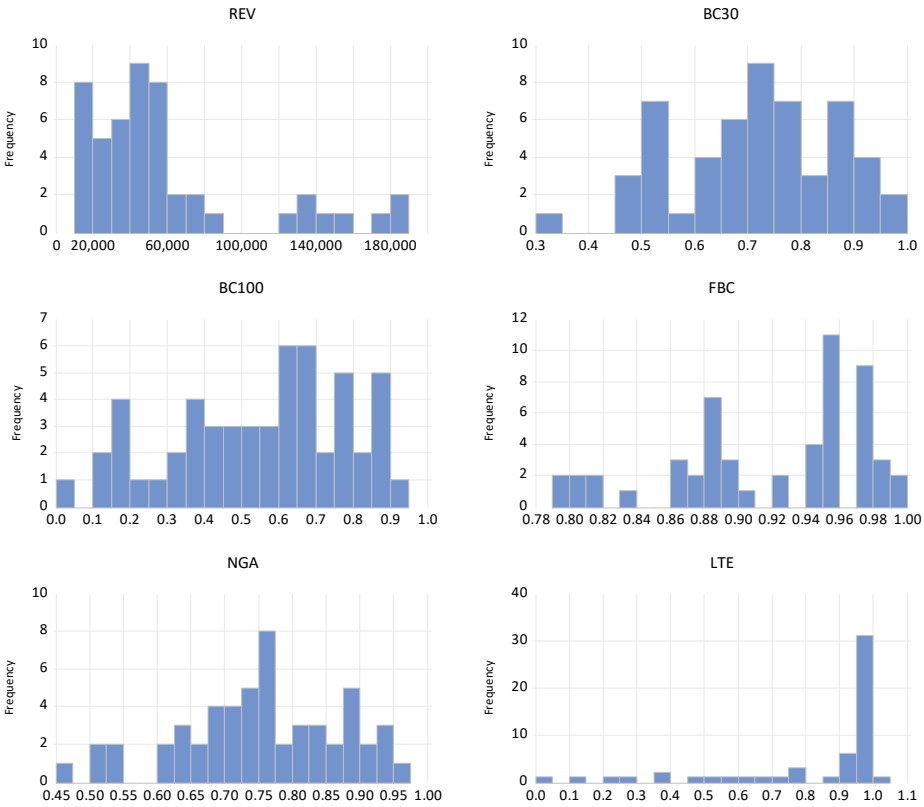

**Figure 4.** Variables' outliers and breakdown points.

To test the stationarity of the data we used LLC unit root test. As we notice from table 4, all data are stationary at level or at first difference according to the p-value associated to t-statistics.

**Table 4.** Unit root test—LLC.

|  | **REV** |  | **BC30** |  | **BC100** |  | **FBC** |  | **NGA** |  | **LTE** |  |
|---|---|---|---|---|---|---|---|---|---|---|---|---|
|  | **level** | **Δ** | **level** | **Δ** | **level** | **Δ** | **level** | **Δ** | **level** | **Δ** | **level** | **Δ** |
| *t*-stat | −0.835 | −4.267 | 0.842 | 1.026 | 4.554 | −8.170 | 5.497 | −0.836 | 5.987 | 1.283 | −35.001 | −78.192 |
| *p*-value | 0.201 | 0.000 | 0.800 | 0.047 | 1.000 | 0.000 | 1.000 | 0.020 | 1.000 | 0.040 | 0.000 | 0.000 |

After examining the stationary properties of the data, the study employs a correlation matrix to confirm that the data for the current investigation is free of the problem of collinearity. Table 5 presents the results of the correlation coefficients, which highlight that there is no multicollinearity problem in the data since the coefficient of correlation among any two variables is less than 1.00.

**Table 5.** Correlation matrix.

|  | **REV** | **BC30** | **BC100** | **FBC** | **NGA** | **LTE** |
|---|---|---|---|---|---|---|
| REV | 1 |  |  |  |  |  |
|  | ----- |  |  |  |  |  |
| BC30 | −0.25382 | 1 |  |  |  |  |
|  | 0.0784 | ----- |  |  |  |  |
| BC100 | −0.1014 | 0.757283 | 1 |  |  |  |
|  | 0.4881 | 0 | ----- |  |  |  |
| FBC | −0.56681 | 0.675601 | 0.384968 | 1 |  |  |
|  | 0 | 0 | 0.0063 | ----- |  |  |

| | | | | | |
|---|---|---|---|---|---|
| NGA | −0.14224 | 0.919367 | 0.73267 | 0.675108 | 1 |
| | 0.3296 | 0 | 0 | 0 | ----- |
| LTE | 0.344341 | 0.516814 | 0.526869 | 0.018028 | 0.506985 | 1 |
| | 0.0154 | 0.0001 | 0.0001 | 0.9022 | 0.0002 | ----- |

In proceeding to apply robust least squares regression on the proposed variables and using the S-estimator as the estimation method, we obtained the results in Table 6.

**Table 6.** Robust least squares regression results for the dependent variable REV.

| Variable | Coefficient | Std. Error | z-Statistic | Prob. |
|---|---|---|---|---|
| BC30 | −83862.42 | 16644.44 | −5.038464 | 0.0000 |
| BC100 | 68263.99 | 6571.827 | 10.38737 | 0.0000 |
| FBC | −8813.001 | 9996.640 | −0.881596 | 0.3780 |
| NGA | 102054.5 | 21841.95 | 4.672409 | 0.0000 |
| LTE | 11596.51 | 4107.112 | 2.823520 | 0.0047 |
| Robust Statistics | | | | |
| R-squared | 0.524693 | Adjusted R-squared | | 0.481483 |
| Scale | 13570.56 | Deviance | | $1.84 \times 10^8$ |
| Rn-squared statistic | 1710.806 | Prob (Rn-squared stat.) | | 0.000000 |

The table reveals the significant impact of broadband speed from a statistical point-of-view, in order that the use of connections with low download speed leads to a reduction in tax revenues, while the use of technological networks with high download speed positively influences the volume of revenues from taxes and fees to the central budgets. The importance of broadband coverage speed can also be explained through the lens of users' behavior; the majority of people are reluctant to use difficult applications which process commands slowly and often encounter blockages during the execution of some commands. This confirms the second hypothesis, H2, regarding the influence of broadband speed coverage on the volume of revenues from taxes and contributions to the central government.

On the one hand, regarding the coverage infrastructure, among the three variables, the FBC variable is statistically insignificant. On the other hand, a strong influence of NGA coverage on the collection of taxes and fees can be observed, as well as the use of mobile broadband which has a positive impact on the payment of taxes to the central institutions. Taking into account that there is at least one mobile connection in every household in the analyzed countries [9], we can observe the significant positive impact of LTE on the growth of revenues from taxes and fees collection. The results in Table 6 confirm hypothesis, H3, regarding the significant positive influence of the degree of broadband coverage on the degree of tax collection.

The bottom portion of the output displays the R-squared and adjusted R-squared and indicates that the model accounts for roughly 50% of the variation in the constant-only model. The Rn-squared statistic of 1710.806 and the corresponding *p*-value of 0.00 indicate a strong rejection of the null hypothesis that all non-intercept coefficients are equal to zero.

To strengthen the results obtained by applying the S-estimator to the proposed model, we additionally performed the Granger causality test [45,50], the results of which are specified in Figure 5.

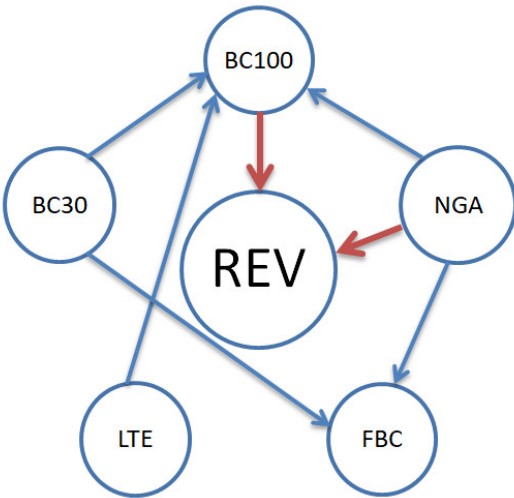

**Figure 5.** Pairwise Granger causality results.

We notice that regarding the direction of causality between the variables, the results of the Granger test indicate a direct causality on the performance of the fiscal policy from the higher speed broadband (BC100), as well as from the modern technological infrastructure (NGA), which can also support higher data transmission speeds.

Causality relationships were also identified between the independent variables: From NGA and LTE to BC100, an absolutely normal relationship, since the development of a modern infrastructure that supports high transmission speeds automatically leads to an increase in speed; from NGA to FBC, also a normal relationship as an increase in new generation infrastructure will lead to the reduction in rudimentary types of infrastructure.

## 5. Conclusions

The paper analyzes the impact of broadband coverage and speed on tax collection in six eastern European countries. The novelty of the study consists in the specific analysis we conducted. To date, no studies have been carried out on the countries included in the analyzed sample to explain the evolution of fiscal policy through the prism of broadband coverage and speed. The analysis is of major importance, especially in the context of the current pandemic, under the pressure of restrictions and fears related to the COVID-19 pandemic, since the foundations of a modern technological infrastructure and the development of digitalization in tax administrations could have a significant positive effect in the protection of citizens. The variables we considered are part of digitalization showing the level of technological development. In the literature review, we could find papers connecting digitalization with economic growth or fiscal performance. However, digitalization was seen more as a general phenomenon rather than being analyzed for specific variables as conducted in this study.

To achieve the proposed objective, three hypotheses were formulated regarding the influence of the technological infrastructure on the level of revenues from taxes and fees at the central governments. The research method used was robust least squares regression with S-estimator, considering the multitude of outliers in the available dataset. The results of the analysis confirmed the three formulated hypotheses, highlighting the importance of the development of the technological infrastructure both in terms of coverage and download speed.

Although at the national level, the technological infrastructure situation is good in at least four of the analyzed states (Bulgaria, the Czech Republic, Hungary, and Romania), it was found that there are sharp discrepancies between the urban and rural environments in all six states [9].

In an era of digitalization combined with the recent pandemic events, decision-makers must take measures to consolidate the technological infrastructure to encourage the large-scale use of the applications made available by tax administrations.

The theoretical implications of our paper reside in the fact that the results of the present study could be used by other researchers and extended to other countries. From a practical point-of-view, the results are helpful for public management, which could develop better strategies for raising the level of taxes collected. Therefore, fiscal policies are interconnected with the ones focused on digitalization, technology, and education.

The limitations of our research refer to the limited time frame for which the data are available. Moreover, there might be important challenges for implementing the results due to the limited financial resources in these countries, bureaucracy or corruption. An inverse relationship of causality from the fiscal policy to the development of the technological infrastructure of a country is present in other studies [51,52], mainly referring to the tax incentives the government could grant to stimulate companies to invest more in their digitalization.

Therefore, for future research directions, we consider extending the panel of countries and creating a cluster analysis considering a comparison between eastern and western European countries. Moreover, the model could be developed to include other variables we noticed in the literature, such as the educational level, the decentralization [20], the corruption index [53], and the relationship between tax incentives and technological development and digitalization [51,52].

**Author Contributions:** Conceptualization, M.D.D. and A.M.Ț.; methodology M.D.D.; software, A.M.Ț.; validation, D.B. and J.P.; formal analysis, M.D.D. and D.B.; investigation, A.M.Ț. and I.P.; resources, I.P.; data curation, J.P. and A.M.Ț.; writing—original draft preparation, I.P. and S.P.; writing—review and editing, S.P. and D.B.; visualization, J.P.; supervision, M.D.D. and S.P.; project administration, D.B. and A.M.Ț. All authors have read and agreed to the published version of the manuscript.

**Funding:** This research received no external funding.

**Conflicts of Interest:** The authors declare no conflict of interest.

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
