# Peer review of "Combining the Broadband Coverage and Speed to Improve Fiscal System Efficiency in the Eastern European Union Countries"

_electronics, doi:10.3390/electronics11203321_

Round 1

Reviewer 1 Report

·         The English language level used in this paper is not up to the standards required for publication in an international research journal. The authors are advised to perform a substantial revision in this regard.

·         A number of abbreviations are used in the paper without using proper terminology to define/describe the physical terms.

·         In the Introduction, the author described past work, but little comment on the contribution and shortcomings. The author needs to provide critical comments.

·         I think, the authors must strengthen the References section with titles that use the same technique and Old references should be replaced by more recent year journal papers.

·         Please highlight how the work advances or increments the field from the present state of knowledge and provide a clear justification for your work.

·         Investigating is too narrow while practical applications may have cons of technology and expenditure involved in this.

Author Response

Dear Reviewer,

Thank you for your observations and for the opportunity to improve our manuscript!

We are very grateful for taking the time to analyze the paper and make very useful, encouraging and thoughtful comments and recommendations.

We have read the evaluation carefully and, based on the review reports, we performed significant revisions of our manuscript, as requested, highlighted with red into the manuscript, respectively:

Response to Reviewer 1 Comments

Point 1: The English language level used in this paper is not up to the standards required for publication in an international research journal. The authors are advised to perform a substantial revision in this regard.

Response 1: We revised the text and made the corrections needed.

Point 2: A number of abbreviations are used in the paper without using proper terminology to define/describe the physical terms.

Response 2: We made the corrections accordingly.

Point 3:   In the Introduction, the author described past work, but little comment on the contribution and shortcomings. The author needs to provide critical comments.

Response 3:  There were added more comments regarding the works mentioned in the Introduction. The paragraph added is highlighted in the text in red.

Point 4:   I think, the authors must strengthen the References section with titles that use the same technique and Old references should be replaced by more recent year journal papers.

Response 4: We made the corrections accordingly. More recent references were added in the paper.

Point 5: Please highlight how the work advances or increments the field from the present state of knowledge and provide a clear justification for your work.

Response 5: We highlighted in the text, in the Conclusion section, the novelty and the contribution and justification of our work.

Point 6: Investigating is too narrow while practical applications may have cons of technology and expenditure involved in this.

Response 6: We added a paragraph in the Conclusions to better highlight the limits and future research directions for our study.

Reviewer 2 Report

The most visible fatal flaw of this work is that it is possible that it is the other way around or that the relationship is bivariate--- those good/strong fiscal policies (Paper’s DV) determines the technological infrastructure of a nation, such as broadband coverage (Paper’s IV1), download speed (Paper’s IV2)—and more likely is the case. The authors did not and should address this very important issue in their paper.

Case in point, developing countries such as the Philippines, Laos, Indonesia, etc. (outside this paper’s sample) with promising fiscal policies tend to perform least on internet speed index references and metrics due to red tape (high yield on corruption index).

There are many factors aside from the two IVs raised that may yield variations in the dependent variable of this study (fiscal policy), but the authors should make sure that the relationship between the variables is accurate/representative of how they behave, even outside the context of this paper’s country samples.

Author Response

Dear Reviewer,

Thank you for your observations and for the opportunity to improve our manuscript!

We are very grateful for taking the time to analyze the paper and make very useful, encouraging and thoughtful comments and recommendations.

We have read the evaluation carefully and, based on the review reports, we performed significant revisions of our manuscript, as requested, highlighted with red into the manuscript, respectively:

Response to Reviewer 2 Comments

Point 1: The most visible fatal flaw of this work is that it is possible that it is the other way around or that the relationship is bivariate--- those good/strong fiscal policies (Paper’s DV) determine the technological infrastructure of a nation, such as broadband coverage (Paper’s IV1), download speed (Paper’s IV2)—and more likely is the case. The authors did not and should address this very important issue in their paper. Case in point, developing countries such as the Philippines, Laos, Indonesia, etc. (outside this paper’s sample) with promising fiscal policies tend to perform least on internet speed index references and metrics due to red tape (high yield on corruption index).

Response 1: To detect the direction of causality between the variables used in this study we applied Granger causality test. The results are stated in the article.  Also, in the Conclusion section, we added a few references to studies that used other variables and mentioned them in the paragraph with future research directions. Among the variables we might use in our future researches, we included also the corruption index which you suggested because it might generate, indeed, interesting results especially in countries with a high index.

Point 2: There are many factors aside from the two IVs raised that may yield variations in the dependent variable of this study (fiscal policy), but the authors should make sure that the relationship between the variables is accurate/representative of how they behave, even outside the context of this paper’s country samples.

Response 2: As we mentioned in the Results and discussions section, in the last paragraph, the selected variables can explain only 50% of the variation of the dependent variable. Obviously, there are many other factors that must be taken into account in the interpretation of the yield of the fiscal policy and which we will take into account in future researches, but the current study refers strictly to the impact of broadband coverage and speed. Although we have not identified works that explain the relationship between the variables included in the model (a relationship that actually explains the novelty of the research), we have mentioned several works that used the same method to perform similar analyses.

Round 2

Reviewer 2 Report

Thank you.